# Perception of Paternal Postpartum Depression among Healthcare Professionals: A Qualitative Study

**DOI:** 10.3390/healthcare12010068

**Published:** 2023-12-28

**Authors:** Aziz Essadek, Alix Marie, Michel-Alexandre Rioux, Emmanuelle Corruble, Florence Gressier

**Affiliations:** 1Interpsy Laboratory, University of Lorraine, 54015 Nancy, France; alix.marie.psychologie@gmail.com (A.M.); michel-alexandre.rioux.cemtl@ssss.gouv.qc.ca (M.-A.R.); 2Centre Intégré Universitaire de Santé et de Services Sociaux (CIUSSS) de l’Est-de-L’Île-de-Montréal, Montréal, QC H1T 2M4, Canada; 3Department of Psychiatry, Bicêtre University Hospital, Assistance Publique Hôpitaux de Paris APHP, University Hospital Paris Saclay, 94275 Le Kremlin Bicêtre, France; emmanuelle.corruble@aphp.fr (E.C.); florence.gressier@aphp.fr (F.G.); 4CESP, INSERM U1018, Moods Team, Faculté de Médecine Paris Saclay, University Paris-Saclay, 94275 Le Kremlin Bicêtre, France

**Keywords:** healthcare professionals, paternal postpartum depression, perception, qualitative study

## Abstract

The pathway to parenthood constitutes a fundamental and transformative stage in every individual’s life. While postpartum depression in mothers has been increasingly studied and acknowledged, paternal postpartum depression (PPD) has garnered only moderate research attention. This study aims to delve into the comprehension and knowledge of healthcare professionals who may encounter men suffering from postpartum depression. Within the framework of this qualitative research, we conducted six semi-structured interviews with various healthcare professionals. The data were subjected to interpretative phenomenological analysis, revealing the following themes: (1) the professionals’ uncertainty in the face of paternal PPD; (2) the context and timing of healthcare professionals’ involvement appeared unsuited for detecting paternal PPD; (3) the experiences of fathers were found not to be shared with healthcare professionals due to their inhibitions and avoidance reactions; (4) the social representation of the role of fathers influenced professionals in their considerations of this aspect. Strengthening the training and confidence of healthcare professionals in France would lead to an enhancement in the screening and management of paternal PPD. Additionally, the healthcare system should better organize postnatal support to enable caregivers to be more available during the peak of depression occurrence.

## 1. Introduction

Postpartum depression (PPD) is commonly associated with women. Over the last decades, research has mainly focused on maternal PPD and child development. However, up to 1 in 10 new dads can experience depression during the perinatal period [1].

Major Depressive Disorder with peripartum onset is defined in the DSM-5 [2] as a subtype of depression that occurs during pregnancy or in the first 4 weeks after delivery. However, symptoms of PPD can occur any time in the first year postpartum. There are not yet diagnostic criteria for paternal PPD. Symptoms described in the existing literature encompass a depressed or sad mood [3]. There may be a loss of interest and changes in appetite or weight. Sleep difficulties and psychomotor agitation or retardation are also reported. Other symptoms include a loss of energy and a diminished ability to think or concentrate. Individuals may experience feelings of worthlessness or excessive guilt, as well as suicidal thoughts. PPD can manifest differently in men. This can be evidenced by higher rates of anger, violent behavior, impulsivity, or risk-taking behaviors [4]. Men may also turn to substances such as alcohol or prescription drugs. Other manifestations include irritability, physical symptoms, social withdrawal, and a significant increase or decrease in working hours [3,4,5].

Paternal PPD was significantly correlated with maternal PPD [1]. Other risk factors have also been identified: past psychiatric episodes, past depression and anxiety [6], variation in hormonal levels such as testosterone, estrogen, cortisol, vasopressin and prolactin [4,5], socio-economic status [6,7], as well as lack of paternal leave [7].

Paternal PPD can significantly impact the father and child relationship and child development, independently of maternal mental health. Depressed fathers tend to perform less positive parenting practices such as reading stories and singing nursery rhymes with their child. Furthermore, depressed fathers are less likely to play outside with their child, thus limiting their discovery and access to the environment [8]. While positive parenting practices (warmth, sensitivity, availability, etc.) decrease in depressed fathers, negative practices (hostile or intrusive behaviors, disengagement, etc.) tend to increase, especially when the mother is also suffering from depression [8].

Several studies have highlighted correlations between PPD and consequences on child development at different ages. In a study conducted 8 weeks after the baby’s birth, Gentile and Fusco highlighted that PPD has a specific and persistent detrimental effect on the early behavioral and emotional development of the child, notably manifested by excessive crying [9]. At 3 years, children of depressed fathers are at risk for hyperactivity and conduct disorder [10]. These findings may be explained by the impact of Paternal Postpartum Depression (PPD) on parent–child interactions [11]. The study by Ramchandani et al. [10] underscores the specific relationship between the behavioral development of a boy and paternal depression. Young boys appear to be particularly sensitive to the effects of paternal parentings [10]. At 6 to 7 years, the risk of developing psychiatric disorders (such as oppositional or conduct disorders) is twice as high in children of depressed fathers than of fathers without depressive symptoms [6]. Paternal perinatal depression has also been correlated to externalized behaviors (aggressive behaviors, conduct disorders, antisocial personality disorder) and internalized behaviors (depression, anxiety, social withdrawal, isolation) in their children during childhood and adolescence [12].

In addition, the risk of adverse development seems to be higher when paternal depression appears in the early months of a child’s life [13]. Fathers can be protective against the development of maternal perinatal mental health problems. Conversely, paternal PPD can negatively impact their partner [14].

So, paternal PPD is a rising concern considering the individual consequences, but also the potential impact on the father–child relationship, child development, partner relationship, mother’s well-being, and the whole family [10,12]. Preventive actions as well as early professional involvement are important because they could support families in overcoming their initial difficulties tied to new parenthood, hopefully leading to a more balanced development of the child. However, fathers seem to receive less support. For example, in a large sample of 499 nurses in Swedish child healthcare, almost 90% of them estimated that it rarely came to their attention that a father was distressed, and less than 20% had offered supportive counseling to any distressed father in the previous year [15]. In addition, fathers described experiences of sometimes being overlooked in contact with the CHC nurse when these ones turned to the mother [16]. In the face of these data, it appears that depressive symptoms must be better screened.

Therefore, medical and mental health professionals (e.g., psychologists, psychiatrists, and midwives) working alongside parents should be able to recognize and identify depressive disorders in future and young fathers.

Paternal PPD seems difficult to detect and only a few studies have focused on healthcare recognition of paternal PPD. In a qualitative descriptive study on Child Health Center nurses’ experiences, paternal PPD was experienced as being vague and difficult to detect [17]. The same study highlights the complexity for professionals to engage with fathers, and when they do interact with them, fathers are often less expressive [17]. Some studies have previously highlighted the difficulty for professionals to identify and treat maternal PPD, partly due to a lack of training [18]. Other research has highlighted professionals’ apprehensions in detecting PPD and the limits of their investment, which can lead to frustration [19]. The risk of underdiagnosis amongst depressed fathers is a lack of proper care, which would lead to increased risks for their own mental health as well as negative impacts on the development of their child. It is therefore important to better understand what hinders the identification of paternal PPD symptoms by healthcare professionals.

The purpose of this study is to explore the key factors that impede the identification of paternal PPD by perinatal healthcare professionals in the postpartum period.

## 2. Method

### 2.1. Approach

This study is part of a phenomenological approach. Phenomenology, a concept introduced by the philosopher Husserl, aims to describe lived experience as it is reported by individuals [20]. The collected experience is derived firsthand from the narratives of participants. Here, the hermeneutical approach converges with the phenomenological. Participants recount their experiences, offering a perspective from which they seek to make subjective sense of their lived and embodied encounters. This intertwining of hermeneutics and phenomenology provides a rich understanding of the nuanced ways in which individuals interpret and navigate their subjective realities. The interviewer then strives to “*make sense of the participant trying to make sense of what is happening to them*” [20], a process which is referred to as a double hermeneutic [20,21]. Thirdly, the idiographic approach consists of examining a specific subject starting from a detailed and singular analysis [20,22].

### 2.2. Participants

In France, after childbirth, parents may encounter several professionals both within the maternity facility and upon returning home. Indeed, midwives, nurses, and psychologists can make home visits to meet with the parents and the infant. These professionals can provide assistance for up to 14 weeks following the child’s birth [23].

The members of the research team approached health professionals from their professional network working in perinatology institutes and likely to meet the selection criteria established in the context of the research. Therefore, this is a convenience sampling. The selection criteria used to recruit potential participants were the following: currently working alongside expecting parents or young parents of children under 12 months of age; regularly and directly involved with expecting or young parents; they have had at least one year of experience working with expecting or young parents. Professionals meeting the selection criteria have been contacted. During this first contact, the objectives of the study, the implications of participation, and the issues of confidentiality were explained. Interested participants gave their consent to participate in writing. Out of about 20 people initially contacted, six professionals agreed to participate in our study.

Table 1 presents the socio-demographic characteristics of the participants. The participants were all women, with a seniority ranging from 1 to 44 years. The distribution of professionals is as follows: a child psychiatrist working in a maternity hospital; a liberal midwife; a liberal midwife specializing in breastfeeding consultation; a midwife in a maternity hospital offering preparation courses for young parents, who recently conducted trials of talk groups aimed exclusively at future fathers; a liberal psychologist working with young parents, especially young mothers; a psychologist in the neonatal department of a maternity hospital.

### 2.3. Interview Guide

The interviews were conducted by the second author at the interviewees’ workplaces in order to accommodate to their schedules. The interviews lasted between 30 min and an hour and were audio-recorded. Each participant was only interviewed once. The semi-structured interview followed a question grid (made up of eight leading questions) that was previously established by the authors based on their extensive literature review [17,18,24,25]. The topics covered are (from the perspective of the participant): the definition, symptoms, etiology, risk factors, interventions recommended, and the impact of paternal PPD, as well as the degree of comfort with this problem.

### 2.4. Procedure

Data transcription. The audio recordings have been subject to a literacy transcription [26], that is to say, a word-for-word transcription taking into account pronunciation deviations.

Interpretative Phenomenological Analysis (IPA). An IPA [20] was conducted on the textual data. The objective when using IPA would thus be the “*detailed examination of personal lived experience, the meaning of experience to participants and how participants make sense of that experience*” [20]. Each of the steps proposed by Smith [20] were carried out, namely: (1) read and reread, (2) initial notations, (3) develop emerging themes, (4) look for links between emerging themes, (5) move on to the next case, and (6) look for patterns between cases. Analyses were supported using Microsoft© Word software (Version 2301).

Credibility check. In order for the data analyses to represent the experiences of the participants, two reflective processes were put in place. First, the research team engaged in a constant reflective process concerning its relationship to the research object. This reflection made it possible to bring to light preconceptions, hitherto ignored by members of the research team, about the French perinatal health system. For example, in the aftermath of the interviews, members of the research team realized that they were biased by an idealized representation of the perinatal care system, in which the father is, in reality, not as involved and present as they imagined. In a second step, in order to avoid that the interpretations are based solely on the subjectivity of a single researcher, the analysis of the data was carried out in a consensual manner. Thus, two coders (two first authors of the article), coded the verbatims independently. Then, during the pooling of the preliminary results, each of the divergent codifications was discussed and debated under the supervision of a researcher external to the codification process (third author of the article), until an agreement was reached. Following these discussion periods, the codifications were adjusted.

## 3. Results

Four main recurring themes were raised by the participants in reference to their experience facing paternal PPD in young fathers. The verbatim extracts were translated from French (the language spoken by the participants) into English by the authors.


**Recognize the symptoms without naming them, the uncertainty of professionals in the face of PPD**


Perinatal healthcare professionals often have accurate representations of paternal PPD and how it is expressed but seem to have a lot of doubt about it. Indeed, our questions about its symptoms, development, and consequences on the family’s well-being were met with a lot of uncertainty and doubt by our participants. Answers such as “*I don’t know*”, “*I can’t give you a very specific answer to that*”, and “*I don’t know if I can be of any real help with that*” were frequently given, even though most of them relayed accurate information when asked about symptoms of depression in young or future fathers. Despite this, the professionals did not feel confident about their knowledge of paternal PPD. This lack of confidence requires a multifaceted perspective on the situation, and even after working in pairs, doubt persisted. “*And, um, yes, when we have doubts, we work together, (...) we do postnatal visits at home, um, and when we ask questions, there you go, it opens up, and we identify the vulnerability of the fathers*”. It seems that perinatal healthcare professionals are able to recognize fathers’ suffering, but they do not use the term of “paternal postpartum depression” to describe their observations, especially those who feel they lack training in psychology or psychiatry. Indeed, professionals seem to think that they do not have enough training in the field to enable them to name PPD: “*Oh yes, I think, I believe I’m missing that, of course. It’s because we’re not adequately trained*”.


**The context and period of perinatal healthcare professionals’ involvement seem unsuitable for detecting paternal postpartum depression**


This second theme was identified based on the challenges that perinatal healthcare professionals face to “meet” future or young fathers in their daily practice. This can be in the physical sense or of building an interpersonal relationship between carer and patient.

First, various professionals explain that they do not meet the fathers at the right time, nor for a sufficient period of time to identify depressive symptoms or tendencies. When thinking about the ideal time to assess the state of the father, Mrs. A., a maternity psychiatrist, referenced the postpartum visit that takes place six weeks after the birth of the baby, during which the midwife meets the parents and the baby to assess their well-being. However, she states that fathers are mostly unavailable by then: “*The men, they don’t come! (…) we don’t see them, the men, they’ve gone back to work by that time*”. Mrs. C., a liberal psychologist, raised the same problem by underlining that young fathers have to go back to work in the weeks following their child’s birth.

Mrs. B., a psychologist in neonatal care, explained that the context of neonatal care might not be appropriated for detecting and caring for depression in young fathers: “*We don’t have time, (…) that time is too peculiar and too permeated by (…) violent, brutal events. The issue of depression cannot be brought up [by the fathers] at this point in time*”. She added, “*The subject has to be able to, um, talk about it enough, right, which often happens in the aftermath*”, suggesting that the psychological elaboration necessary to process the new paternity (with the added complications tied to neonatal care), could only take place after the birth of the baby. Mrs. D., a maternity midwife, shared Mrs. B.’s views on the difficulties of recognizing distress in young fathers, even in the first few days after birth. “*In the immediate postpartum, (…) when I see the couples, they are fulfilled, so it might not be enough to appreciate if there are depressive tendencies or not*. *(…) We don’t offer a follow-up, um, of care, (…) the father, there is no monitoring for him*”. Mrs. D. deplored that young fathers seem to be left to themselves to handle their new paternity, without any real professional support. Another midwife working as a liberal, Mrs. E., denounced the lack of psychological support for future and young fathers during conventional pregnancies: “*It’s true that we only take the father into account when there are, um, dramatic events*”. A position supported by Mrs. H. (maternity psychiatrist), who stated that she mostly meets fathers “*during prenatal diagnostics, for surgical interventions, during grief…*”. These statements suggest that fathers are mostly considered by perinatal mental healthcare if dramatic events occur. Mrs. F., a liberal midwife who formerly worked in maternities, explained that context might also play a role in the care directed to the father. She stated: “*In maternities, we’re focused on the mother and the baby*”, as opposed to her own independent work, where home visits made her feel closer to the families and to the fathers.


**Fathers’ experiences are not shared with perinatal healthcare professionals due to their inhibitions and “flight” reactions**


The next identified theme revolves around a representation shared by most participants that men have greater inhibitions when expressing themselves and their emotions than women. “*It’s complicated to target them, with the ‘male pride’, um, they’re not going to tell that they’re feeling depressed*”, Mrs. D. (maternity midwife) said about men. She added, “*They don’t speak up*”. On multiple occasions, it seems that mothers can act as a channel through which the fathers’ distress is addressed by professionals. Mrs. C., (liberal psychologist), meets very few young fathers struggling with their new paternity, but stated: “*My patients are mostly women seeking counseling, um, sometimes when they mention their spouse’s difficulties, they ask if their spouse could come with them to appointments*”. Mrs. A. (maternity psychiatrist) also remarked: “*Ultimately, all of the men that I met were sent by their wives*”. Remembering a particular patient, she added that: “*He didn’t want to come and talk on his own, and in the end, he comes and talks with his wife. His wife helps him to talk about it*”.

While some participants described fathers that were inhibited or withdrawn from the carer–patient relationship, others mentioned the “flight” reaction that they sometimes observe. When reflecting on the scarcity of depressed fathers that she met during her practice, Mrs. A. (maternity psychiatrist) assumes that “*They tend to run away. Maybe that’s why we don’t see them. Maybe it’s because depressed fathers, well, they’ve left. Because there are quite a few that leave their wife during the pregnancy. Maybe that is how it [depression] manifests itself*”. Mrs. D. (maternity midwife) and Mrs. F. both use terms like “*flight*”, “*running away*”, “*walking away from everyone (…), wanting to take a step back, even though it was a planned pregnancy (…)*” as symptoms that they would associate with paternal postpartum depression. Mrs. A. (maternity psychiatrist) considers it not only a symptom, but a possible defensive act against paternal depression: “*I think many men leave when they are depressed. I think they find these ways of defending themselves, by acting out somehow*”. According to Mrs. A., running away could allow the father to distance himself from his own suffering.


**The social representation of the place of fathers influences professionals in their consideration of them.**


The fourth theme explores the evolution of societal norms surrounding the representations of fathers’ function and role in a family and the challenges that these representations pose to young fathers discovering their new fatherhood. In this sense, social norms will influence professionals who may not perceive the specific situations of fathers. Indeed, participants felt that society’s expectations of a father are evolving and that identifying with their fathers in order to define their own fatherhood is increasingly harder: “*I often hear men say ‘in my father’s time, his role wasn’t the same (…) I can’t rely on what he did’, because that is not what is asked of dads today*” (Mrs. C., liberal psychologist). Mrs. E. (liberal midwife) added that “*We ask them to get involved more, to be way more present as a partner and as a dad, but on the other hand, (…) we don’t really pay attention to their experience*”.

Furthermore, it is acknowledged that some men struggle with reconciling their position as a man and partner with their new fatherhood: “*There are some concerns about, um, ‘what kind of husband am I going to be?’ or ‘I’m being asked to be paternal, but won’t I lose my manhood? (…) Won’t I lose my status as a husband by becoming a father?’”* (Mrs. C., liberal psychologist). These questions refer to a real shift in the man’s identity that occurs when he becomes a father, which might be difficult to navigate, as Mrs. B. explains: “*We tend to forget about this intrinsic masculine position of the man who becomes a father (…) It raises the question of becoming a father while remaining a man*”.

Professionals emphasize the importance of the transformation of the paternal position and the complexity for fathers to cope with these changes, but above all for them, the professionals, to find their way through these changes.

## 4. Discussion

To our knowledge, this is the first study on perinatal health professionals and their perception of paternal PPD. Our results shed light on different potential factors that may contribute to hinder the acknowledgement of paternal PPD by professionals working with expectant families during the perinatal period.

First, the participants reported feeling a great deal of doubt about the diagnosis of paternal PPD despite knowing about the disorder and having accurate representations of how its symptoms manifest. This result partially aligns with the findings of Santos Junior et al. [18], which highlighted the varying levels of professional training that could lead to the issue being overlooked by some professionals. However, the study by Santos Junior et al. [18] focused on maternal PPD, which is more extensively documented compared to that of fathers. Additionally, this diagnostic difficulty echoes the experience of fathers with PPD symptoms. Indeed, these fathers would be able to perceive their behavioral changes, but would tend to normalize them rather than attribute them to a PPD [27]. This could be due to a lack of information available for fathers [27,28], as well as the fact that PPD is considered as a female mental disorder [27]. According to Pederson et al. [27], these last elements are likely to prevent fathers from seeking help from health professionals.

Second, the participants identified a difficulty in accessing future fathers, whether on a physical or relational level. According to our participants’ testimonies, fathers have very few interactions with the perinatal care workers. In fact, they would be mostly involved in the care process under specific circumstances such as medically assisted procreation, or in dramatic situations such as perinatal loss. These contexts remain exceptional and do not reflect the usual trajectory of young fathers susceptible to developing postpartum depression. In the absence of the father, the mother, therefore, becomes the only gateway to the father’s mental health, which greatly complicates the identification of paternal PPD symptoms. This difficulty surrounding direct access to fathers was also noted by Hammarlund et al. [17]. One explanation is the duration of paternity leave, even though it was doubled in France in 2021, from 14 to 28 days, seven of which are mandatory, which would not ensure the presence of the father at the various follow-ups. Another explanation is that fathers can sometimes feel excluded from medical consultations when the attention is rather on the mother and the child [16].

Thirdly, the participants identified a difficulty for future fathers to open up about their experiences and therefore about their symptoms. This difficulty for fathers to talk about their emotions was also reported by participants in the study of Hammarlund et al. (2015). One potential explanation is that fathers with symptoms of PPD report feeling that talking about their distress is taboo [27,29]. These fathers reported feeling being judged or stigmatized if they open up [27]. They can feel that they are not able to meet the standards of masculinity, for example, that they should be strong enough to handle the situation [28] or that their experience is not legitimate, for example, considering that it is the mother who spends the vast majority of her time with their child and that in this sense, they should be able to manage [28]. This point appears to be crucial, as a recent study highlighted the risk of PPD among fathers who do not take their paternity leave [30]. The reluctance to take paternity leave can be influenced by representations of masculinity norms and may increase the risk of PPD.

Finally, the participants reported that the significant social changes regarding the father’s role and his place in the family can lead to a significant psychological reorganization in fathers. The participants (all women) seem to perceive its contours but not its full scope, which gives them the impression of not being able to consider the experience of fathers in all its complexity. Social changes considering the father’s place in the family system can generate among them a feeling of loss of control and helplessness. In fact, men report receiving mixed messages about equality in terms of parenthood: On the one hand, society asks them to be equal to the mother, and on the other hand, this same society considers the mother as the primary parent [31]. These social changes are a challenge for both fathers and health professionals. These results also resonate with the findings of Holopainen et al. [24], which emphasized the complexity of perceiving the various changes in fathers with PPD.

## 5. Limits

Our work has some limitations. First, our sample size cannot represent all perinatal care workers. Only three different professions were represented, and only women were interviewed. The inclusion of male participants, especially if they were fathers themselves, could lead to slightly different results: men could be more sensitive to the social and psychological place left to the father and thus be more sensitive to symptoms. However, in France, most perinatal caregivers are women. For example, 2.6% of midwives are men in France [32]. Enlisting professionals from different disciplines can introduce variations in training and consequently in their comprehension of paternal PPD. However, the father’s mental health in the perinatal period remains poorly investigated, regardless of all professionals. However, the recruited professionals belong to the same network and are thus likely to meet regularly within the framework of inter-institutional care. Second, this study is limited to the French context, involving representations and social conditions specific to this country surrounding the question of the father and mental health. The results of this study must therefore be placed in this specific context. Additionally, the transferability of its results depends on a good understanding of these contextual issues.

## 6. Conclusions

Our results highlight the importance of providing training to all perinatal health professionals on paternal PPD. Screening tools, such as the Edinburgh Postnatal Depression Scale (EPDS), which has been validated also for fathers, can help with the diagnosis that is primarily a clinical one [33]. The EPDS-Partner could also be used to detect paternal PPD through the data reported by the mother [34]. Although it was not raised by the people interviewed, the later peak of depression could be an obstacle to early screening. Indeed, the peak for paternal PPD was reported between 3 and 6 months after birth (25.6%) and the lowest rate during the first 3 months (7.7%) [1]. It is also important to inform the fathers about PPD by providing clear information so that they can turn to the professionals in case of possible mood changes or other behavioral changes. With regard to the difficulties in having access to fathers, it would be important to provide social conditions allowing fathers to be present (e.g., appointments on behalf of both parents, longer parental leave). It is important that they feel included as much as mothers and therefore are addressed during medical appointments. Social and emotional support is necessary [35]. Interventions that support the couple and strengthen their relationship could be beneficial. The quality of the couple’s exchanges, the mother’s encouragement, and joint preparation for the baby’s birth could help the father [36]. The couple’s alliance allows the father to feel less excluded from the dyad and prevents him from feeling jealous of the child. Quality family support can prevent paternal PPD. Fathers’ mental health should not be neglected, both during their wife’s pregnancy and in the postpartum period. Fathers suffering from PPD must be treated for himself, his partner, and the child [35].

The support that society can provide, such as paternity leave, will enable fathers to adapt better to the change in their lives. Worldwide, many countries, including France, offer fathers the possibility of paternity leave. The longer fathers take paternity leave, the better their attitude to caring for their child.

Nonetheless, the possibilities for helping fathers are still limited. Childbirth preparation sessions should be open to men, as should postpartum therapies. Specialized talks and discussion groups for fathers only are beginning to be offered.

Finally, regarding the tendency of fathers not to share their emotional experience, it would be important to raise awareness among the general public to paternal PPD in order to validate and legitimize the experience of fathers who suffer from it. Barriers related to potential stereotypes could be partially removed. In addition, it would be important for health professionals to openly discuss this condition with fathers and explicitly state that it is not taboo, that they are present to hear, and that it does not make them less masculine. Similarly, communication tools such as posters or leaflets to disseminate information about paternal PPD and to raise awareness among a wide audience should be developed. These could be useful in doctors’ waiting rooms, maternal and child protection centers, maternity wards, and pharmacies.

Encouraging fathers to take part in their baby’s care is primordial in allowing fathers to take their place. It can also be used to assess fathers’ abilities and to detect an early paternal PPD episode. Indeed, treating parents at an early stage, as soon as the first symptoms such as anxiety and/or sadness appear, would make it possible to reduce the impact on the child’s psychomotor development.

## 7. Avenues for Future Research

Our results suggest avenues for future research. First, it would be interesting to interview male health professionals to see if there are differences in their experience compared to their female colleagues. In addition, it would be interesting to evaluate the effect of training on paternal PPD and the implementation of a screening procedure using systematic tools on the ease of professionals in identifying the symptoms of paternal PPD and diagnose it.

## Figures and Tables

**Table 1 healthcare-12-00068-t001:** Socio-demographic characteristics of study participants.

Participants	Occupations	Workplaces	Specializations	Seniority
Mrs. A.	Child psychiatrist	Maternity hospital	Perinatal	23 years
Mrs. B.	Psychologist	Maternity hospital	neonatal	30 years
Mrs. C.	Psychologist	Liberal	Young parents	14 years
Mrs. D.	Midwife	Maternity hospital	Preparation courses for young parents	44 years
Mrs. E.	Midwife	Liberal	Young parents Breastfeeding consultation	2 years
Mrs. F.	Midwife	Liberal	Breastfeeding consultation	11 years

## Data Availability

The datasets used and/or analyzed during the current study are available from the corresponding author upon reasonable request.

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
