# Peer review of "Perception of Paternal Postpartum Depression among Healthcare Professionals: A Qualitative Study"

_healthcare, 2023, doi:10.3390/healthcare12010068_

Round 1

Reviewer 1 Report

Comments and Suggestions for Authors

The review of the manuscript entitled: “Perception of Paternal Postpartum Depression Among Healthcare Professionals: A Qualitative Study”

The study is a qualitative study, which aimed to explore the key factors that impede the identification of paternal PPD by perinatal health-care professionals in the post-partum period. For this purpose authors recruited six professionals who were currently working alongside expecting or young parents and made a semi-structured interview with them.

Comments for Authors:

Thank you for the research you have done. The topic is interesting. However, the language of the paper needs to be edited. It is recommended that authors break long boring sentences to multiple short sentences and simplify sentences which are difficult to understand by using more convenient and usual words and phrases. Furthermore, there are some typing errors in the text (e.g. in line 276, “the” is wrongly repeated in the sentence).

Good luck

Comments on the Quality of English Language

It is recommended that authors break long boring sentences to multiple short sentences and simplify sentences which are difficult to understand by using more convenient and usual words and phrases.

Author Response

Reviewer 1:

The review of the manuscript entitled: “Perception of Paternal Postpartum Depression Among Healthcare Professionals: A Qualitative Study”

Comments for Authors:

Thank you for the research you have done. The topic is interesting. However, the language of the paper needs to be edited. It is recommended that authors break long boring sentences to multiple short sentences and simplify sentences which are difficult to understand by using more convenient and usual words and phrases.

Response: We would like to thank the Reviewer for his careful reading. We have made changes throughout the text, reducing long sentences and modifying certain terms. A number of errors have also been identified and corrected.

Furthermore, there are some typing errors in the text (e.g. in line 276, “the” is wrongly repeated in the sentence).

Response: Thanks for proofreading, this error has been removed.

Reviewer 2 Report

Comments and Suggestions for Authors

Thank you for helping to bring awareness to this important area of mental health. 

This manuscript presents themes that emerged from a qualitative study of six health care professionals' experiences related to identifying paternal postpartum depression.  The overall structure, design, and description of the study is well done.  A couple minor recommendations are made for improvement:

1.  The introduction includes a description of the adverse childhood effects of paternal PPD.  That information should include more transitions and explanations.  For example, the bold claims of paternal PPD leading to conduct disorder should be explained in more detail, with more transitional statements showing the importance of identifying paternal PPD.

2.  The results section details the various themes detected through the interviews.  The first theme should be elaborated more; more detail and example quotations should be provided.

Author Response

Reviewer 2:

Thank you for helping to bring awareness to this important area of mental health. 

This manuscript presents themes that emerged from a qualitative study of six health care professionals' experiences related to identifying paternal postpartum depression.  The overall structure, design, and description of the study is well done.  A couple minor recommendations are made for improvement:

  1. The introduction includes a description of the adverse childhood effects of paternal PPD.  That information should include more transitions and explanations.  For example, the bold claims of paternal PPD leading to conduct disorder should be explained in more detail, with more transitional statements showing the importance of identifying paternal PPD.

Response : We thank the reviewer for these relevant comments. We have partially rewritten this paragraph and added references to support our statements.

Change In text :

Several studies have highlighted correlations between PPD and consequences on child development at different ages. In a study conducted 8 weeks after the baby's birth, Gentile and Fusco highlighted that PPD has a specific and persistent detrimental effect on the early behavioral and emotional development of the child, notably manifested by excessive crying [9]. At 3 years, children of depressed fathers are at risk for hyperactivity and conduct disorder [10]. These findings may be explained by the impact of Paternal Postpartum Depression (PPD) on parent-child interactions[11]. The study by Ramchandani et al. [10]underscores the specific relationship between the behavioral development of a boy and paternal depression. Young boys appear to be particularly sensitive to the effects of paternal parentings[10].

  1. The results section details the various themes detected through the interviews.  The first theme should be elaborated more; more detail and example quotations should be provided.

Response : We thank you for your expertise that allowed us to review the text. In order to provide more substantiation, we have added the following elements:

Change In text :

This lack of confidence requires a multifaceted perspective on the situation, and even after working in pairs, doubt persisted. "And, um, yes, when we have doubts, we work together, (...) we do postnatal visits at home, um, and when we ask questions, there you go, it opens up, and we identify the vulnerability of the fathers."

those who feel they lack training in psychology or psychiatry. Indeed, professionals seem to think that they don't have enough training in the field to enable them to name PPD; "Oh yes, I think, I believe I'm missing that, of course. It's because we're not adequately trained”.

Reviewer 3 Report

Comments and Suggestions for Authors

Thank you for the great opportunity to review your valuable research. The current study showed that to perceive paternal postpartum depression among healthcare professionals in France using a phenomenological analysis. The study’s originality and subject matter are a high novel. However, there are several ambiguities in the methods and interpretation. Moreover, I believe there is a lack of emphasis on certain aspects of medicine and health care research. While the authors explained the parental PPD, the methods and results lacked sufficient evidence to support these relationships. Major or minor comments are detailed as follows.

Page 3, line 132 in Table 1.

It needs to include more information about parental or maternal because they are parties who directly or indirectly suffer from parental PPD. This may include parental age, employment status, the number of children, years of marriage, etc. Therefore, it is necessary to provide more information not only about professional but also about their customers or service recipients. Additionally, more information about professionals, such as professionals age, their areas of active in providing services, and their year of experiences, should be included. If possible, please add more information about both professionals and parents.

Page 7, line 330

The authors mentioned that the sample size cannot represent all perinatal care workers. I completely agree with this statement, and this is severe limitation of this work. In fact, the readers don’t know the professional’s activity region, there experience, etc. This could introduce bias into the results of the study. However, the authors explain that most of perinatal care givers are women. I believe this cannot be a valid defense for this limitation. Moreover, the professionals had limited opportunity to meet the future father, which means they did not thoroughly investigate and provide opinions on parental PPD. While I understand this is too limited subject of research, I am not sure this evidence can adequately explain for parental PPD.

Page 7, line 345-346

Please add the abbreviation of Edinburgh Postnatal Depression Scale (EPDS) upon its first use in the manuscript.

Page 8, line 347

There is no further use of EPDS-P after this line, so please remove this abbreviation.

Page 8, line 357-364

Please support the authors’ opinion with references. There is no citation providing supporting evidence.

Comments on the Quality of English Language

 Minor editing of English language required

Author Response

Reviwer 3:

Thank you for the great opportunity to review your valuable research. The current study showed that to perceive paternal postpartum depression among healthcare professionals in France using a phenomenological analysis. The study’s originality and subject matter are a high novel. However, there are several ambiguities in the methods and interpretation. Moreover, I believe there is a lack of emphasis on certain aspects of medicine and health care research. While the authors explained the parental PPD, the methods and results lacked sufficient evidence to support these relationships. Major or minor comments are detailed as follows.

Response : Dear reviewer, we would like to express our gratitude for the thorough reading you have conducted on our manuscript, and we appreciate the support you provide to this research.

Page 3, line 132 in Table 1.

It needs to include more information about parental or maternal because they are parties who directly or indirectly suffer from parental PPD. This may include parental age, employment status, the number of children, years of marriage, etc. Therefore, it is necessary to provide more information not only about professional but also about their customers or service recipients. Additionally, more information about professionals, such as professionals age, their areas of active in providing services, and their year of experiences, should be included. If possible, please add more information about both professionals and parents.

Response : Thank you for this comment. We have added a column in the table indicating the seniority of the professionals we interviewed. However, we are not in a position to present precise information on parents. We have some information on the elements communicated by the professionals. The profile of parents described corresponds to young parents, for whom this is their first child. The research focuses on professionals in contact with parents of newborn infants. The selection criteria were defined as mentioned on page 3: The selection criteria used to recruit potential participants were the following: currently working alongside expecting parents or young parents of children under 12 months of age; regularly and directly involved with expecting or young parents; they have had at least one year of experience working with expecting or young parents.

Page 7, line 330

The authors mentioned that the sample size cannot represent all perinatal care workers. I completely agree with this statement, and this is severe limitation of this work. In fact, the readers don’t know the professional’s activity region, there experience, etc. This could introduce bias into the results of the study. However, the authors explain that most of perinatal care givers are women. I believe this cannot be a valid defense for this limitation. Moreover, the professionals had limited opportunity to meet the future father, which means they did not thoroughly investigate and provide opinions on parental PPD. While I understand this is too limited subject of research, I am not sure this evidence can adequately explain for parental PPD. 

Response : Dear reviewer, we acknowledge the limitations you have highlighted. In order to provide further justification, we have added the following sentences along with a reference.

For example, 2.6% of midwives are men in France. [32]

  1. Le Dû, M. Synthèse Entre Cure et Care : Les Sages-Femmes Déboussolent Le Genre. clio 2019, 137–151, doi:10.4000/clio.16300.

However, the recruited professionals belong to the same network and are thus likely to meet regularly within the framework of inter-institutional care

 Page 7, line 345-346

Please add the abbreviation of Edinburgh Postnatal Depression Scale (EPDS) upon its first use in the manuscript.

Response : Thank you for your careful reading; we have added this element.

Page 8, line 347

There is no further use of EPDS-P after this line, so please remove this abbreviation.

Response : Thank you for your careful reading; we removed this abbreviation

Page 8, line 357-364

Please support the authors’ opinion with references. There is no citation providing supporting evidence.

Response : Thank you for giving us the opportunity to justify these elements. We have added references 35 and 36 to the text. They have also been included in the reference section.

  1. Gressier, F.; Tabat-Bouher, M.; Cazas, O.; Hardy, P. [Paternal postpartum depression: a review]. Presse Med 2015, 44, 418–424, doi:10.1016/j.lpm.2014.09.022.
  2. Bruno, A.; Celebre, L.; Mento, C.; Rizzo, A.; Silvestri, M.C.; De Stefano, R.; Zoccali, R.A.; Muscatello, M.R.A. When Fathers Begin to Falter: A Comprehensive Review on Paternal Perinatal Depression. Int J Environ Res Public Health 2020, 17, 1139, doi:10.3390/ijerph17041139.